

# Effects of probiotic supplements on growth performance and intestinal microbiota of partridge shank broiler chicks

Yizhe Ye[1,2,*], Zhiquan Li[1,2,*], Ping Wang[1], Bin Zhu[3], Min Zhao[1,2], Dongyan Huang[1,2], Yu Ye[1,2], Zhen Ding[1,2], Longrui Li[4], Gen Wan[1,2], Qiong Wu[1,2], Deping Song[1,2] and Yuxin Tang[1,2]

[1] Department of Preventive Veterinary Medicine, College of Animal Science and Technology, Jiangxi Agricultural University, Nanchang, China
[2] Jiangxi Engineering Research Center for Animal Health Products, Jiangxi Agricultural University, Nanchang, China
[3] Jiangxi Red Animal Health Products Co., LTD., Nanchang, China
[4] Jiangxi Newtoldhow Animal Pharmaceutical Co., LTD, Ji'an, China
[*] These authors contributed equally to this work.

Corresponding author
Deping Song, sdp8701@jxau.edu.cn

## ABSTRACT

**Background.** The benefits of probiotics being used in animals are well-documented via evidenced growth performance improvement and positive modulations of gut microbiota (GM). Thus, a combination of effective microorganisms (EM) has been frequently used in animal production, including broilers. However, there are only very limited reports of EM on the growth performance and the modulation in GM of partridge shank broiler chicks.

**Methods.** We attempted to evaluate the effects of a basal diet with the addition of an EM mixture on the growth performance and gut microbiome of the chicks. A total of 100 ten-day-old female partridge shank broiler chicks were randomly divided into two groups of 50 chicks each, of which, one group fed with EM supplementation in the basal diet (designated as EM-treated group), the other group just fed with a basal diet (referred as to non-EM treated group or control group). The body weight, daily feed intake, daily gain, feed conversion ratio and other growth parameters were observed and compared between EM-treated and non-EM-treated chicks, and the gut microbiota was profiled by 16S rRNA-based next generation sequencing (NGS).

**Results.** EM-treated chicks showed significantly increased performances in body weight (BW), average daily gain (ADG) and reduced feed conversion ratio (FCR). Histological observation indicated that dietary supplementation of EM significantly increased the villus heights (VH) and the ratio of villus height to crypt depth (VH/CD), while decreased the CD of jejunum, ilea, and ceca. The results of 16S rRNA-based gut microbiota analyses showed that *Firmicutes* accounted for the most of the relative abundance (63.24%~92.63%), followed by *Proteobacteria* (0.62%~23.94%), *Bacteroidetes* (0.80%~7.85%), *Actinobacteria* (0.06%~13.69%) and others in both EM-treated and non-EM-treated broiler chicks. The addition of EM could not alter the alpha diversity of gut microbiota. Compared with the non-EM-treated chicks, the abundances of bad bacteria in the phyla of *Firmicutes*, *Euryarchaeota*, and *Ruminococcus* were dramatically

decreased in that of EM-treated chicks, while the abundances of good bacteria in the phyla of *Actinobacteria* and *WPS-2* were significantly increased.

**Conclusions**. The supplementation of EM in feed could improve the growth performance and positively influence the morphological characteristics of the intestine, and ameliorate the community and structure of the intestinal microbiota of partridge shank broiler chicks.

# INTRODUCTION

Feed cost accounts for about 70%∼80% of the total cost of poultry production. Thus, great efforts have been made to improve the nutritive values of feeds to enhance growth performance and health of animals (*Ahmad et al., 2018*). Probiotics, defined as ''live microorganisms'', are one of the major feed additives routinely being used in animal production for decades due to the conferred health benefits to the host when administered in an adequate amount (*FAO, 2002*; *Markowiak & Liewska, 2018*; *Iriti et al., 2019*; *Reszka et al., 2020*). For poultry, probiotics could improve feed intake and digestion efficiency by increasing the activity of digestive enzymes, keep the balance of bacteria in gastrointestinal (GI) tract, promote the gut integrity and thus improve the growth performance and health of birds (*Johnson et al., 2018*; *Soomro et al., 2019*; *Hack et al., 2020*). *Patidar & Prajapati (1999)* showed that *Lactobacilli* supplementation increased the titers of hemagglutination inhibition antibodies of chicks after feeding for 3–4 weeks. *Vinayasree et al. (2012)* evaluated the effect of probiotic organisms on the performance of broilers, and found with the use of probiotics fecal coliform bacteria counted at the end of 6th week in experiment group were significantly lower when compared to the control groups. *Fazelnia et al. (2021)* showed that the dietary supplementation of potential probiotics *Bacillus subtilis*, *Bacillus licheniformis*, and *Saccharomyces cerevisiae* and synbiotic improved the growth performance and immune responses of broiler chicks. Additionally, supplementation of synbiotic and probiotic alleviated the negative effects of *S. typhimurium* on growth and immunity of broiler chicks. Also, other studies have reported the beneficial effects of different probiotics on the broiler growth (*Zuanon et al., 1998*; *Ergun, Yalcin & Sacakli, 2000*; *Vicente et al., 2007*; *Rehman et al., 2020*).

In 1991, Terou Higa reported a multifunctional microbe flora composed of more than 80 kinds of microorganisms, named as Effective Microorganisms (EM) (*Higa, 1991*; *Aruoma et al., 2002*). The dominant bacteria in the EM are *Lactobacillus*, photosynthetic bacteria, *Actinomycetes*, yeasts and filamentous bacteria. Nowadays, EM has been widely used in more than 40 countries and/or areas, including Japan, the United States, India and China (*Rybarczyk et al., 2016*; *Li et al., 1994*). Previous studies demonstrated that EM can improve soil performance, promote crop growth and enhance plant stress resistance. Investigations on broilers carried out by *Chantsawang & Watcharangkul (1999)* showed

that EM could increase body weight, feed intake, feed conversion efficiency, and immune response of chicks. *Safalaoh (2006)* conducted a study on the effect of EM on body weight gain, dressing percentage, abdominal fat and serum cholesterol content of broilers by supplementing EM in drinking water, and it was found that birds fed with EM had higher weight gains, feed efficiency, while lower feed intake and serum cholesterol content than that in control birds. Further studies showed that EM could also improve meat quality, increase slaughter rate, and reduce the rate of death in economic animals (*Jagdish & Sen, 1993*; *Alvarez, Barrera & Gonzalez, 1994*; *Silva et al., 2000*; *Patterson & Burkholder, 2003*; *Alagawany et al., 2018*; *Abd et al., 2020*). In contrast, some research results reported that the supplementation of EM in chicken's feed had no significant effect on mortality, feed conversion ratio (FCR) and weight gain (*Wondmeneh, Getachew & Dessie, 2011*).

Partridge shank chick, a local broiler breed in China, is a relatively smaller body size chick with the features of tender meat and high nutritional value as well as a special flavor when cooked. Therefore, it is very well favored by consumers in China. According to the previous reports, local breeds of broiler chick account for 46.52% of broiler meat in China in 2017, and this proportion was continuously increased in 2018 (*Zhao et al., 2019*). However, the growth rate of partridge shank broiler chick is slow. This characteristic may attribute to its genetic basis, the environmental factor(s), nutrition, and so on. The gut of animals is important site of nutrient absorption in animals, and better development of the intestinal system could benefit the nutrient absorption and improve animal growth performance and health (*Mekbungwan, Yamauchi & Sakaida, 2004*). However, the information regarding the development of intestinal histomorphology and gut microbiota of partridge shank broiler chicks is roughly unknown, and so does the knowledge on the effect of EM on the growth performance of the broilers. We hypothesized that the EM would improve the growth performance and the structure and composition of gut microbiota, perhaps *via* a mechanism of inhibiting the colonization of bad bacteria. The aim of the study was to evaluate the effects of EM on the growth performance, intestinal gut health and gut microbiota of partridge shank broilers.

## MATERIALS AND METHODS

### Ethics statement

All procedures involving live animals were verified and approved by the Office of Animal Care and Use of Jiangxi Agricultural University (protocol number JXAU-LL-20190022). The chicks used in this study were housed at the Animal Research Unit of Jiangxi Agricultural University, located at the college of Animal Science and Technology in Nanchang, Jiangxi, China.

### Chicks and experimental design

Ten-day-old female partridge shank broiler chicks ($n = 100$) were purchased from a local commercial hatchery. All of the chicks had been individually wing-tagged, and immunized with the vaccines against Marek's disease, Newcastle disease, and Infectious bursal disease at 1, 4 and 10 days old, respectively. The experiment was carried out for a 20-day period starting in Jan, 2020. All broiler chicks had ad libitum access to feed and water, and the feed

was offered four times daily at 06.00 am, 11.00 am, 16.00 pm, and 20.00 pm, respectively. The chicks were then randomly divided into two experimental groups, and each group included 5 repetitions with 10 chicks per replication. All chicks of each replication were housed in $0.96 \times 0.96$ m chick coops (at the Chicken Experimental Unit, no. 109, Jiangxi Agricultural University, China) under the same environmental conditions, including a constant temperature of 28 to 31 °C and 20 h daily lighting access throughout the experiment.

The nutrient levels of the basal diet (maize-soybean-based meal diet) corresponded to the NRC (1994) recommended requirements for broilers (Table 1). The EM suspensions used in this study was generously provided by Prof. Nanhui Chen (Department of Preventive Veterinary Medicine, Jiangxi Agricultural University, China). Chickens in experimental group, designated as EM-treated group (abbreviated as group EM), were fed a basal diet supplemented with 0.5 ml (about $2.5 \times 10^9$ colony-forming unit) EM per chick/day for 20 days and the chicks in the negative control group, designated as non-EM-treated group (abbreviated as group B) were just fed by the basal diet for 20 days. The bacterial composition of the EM used in this study was determined by 16S rRNA sequencing on Illumina HiSeq 4000 platform, and the bacterial background information of EM used is supplied in Table S1. The initial and final weights, daily feed intake of the chicks in each group were recorded, and the feces of five chicks in each group were sampled at the 1st, 10th, and 20th experimental day. Then the feed intake was daily measured, body weight (BW) gain was measured at the end of experiment and then the data were used to calculate average daily intake (ADFI), average daily gain (ADG), and feed/gain ratio (F/G). At the 20th experimental day, five chicks of each group were euthanized by pentobarbital sodium and dissected, and the intestinal tissues and cecal contents were collected. For gut microbiota profiling, excreta at the 1st, 10th and 20th experimental day and cecal contents at the 20th experimental day were collected in both groups (designated as EM0, EM10, EM20, and EM20C for samples from EM-treated chicks; and B0, B10, B20, and B20C for samples from non-EM-treated chicks).

## Histomorphology observation

At necropsy, different sections of intestines were examined and collected for histological observation according the previous methods in our lab (Zhang et al., 2020). For each tissue section, at least ten villi and crypts were measured using the cellSens Standard system (Olympus, Japan) with villous height (VH) and crypt depth (CD), which would be used for the calculation of VH/CD ratio.

## Bacterial DNA extraction and 16S rRNA gene sequencing

Bacterial genomic DNA were extracted by the QIAamp DNA Stool Mini Kit (Qiagen, Hilden, Germany) and quantified according to the previous method (Song et al., 2017). Amplification of the hypervariable V4 region of 16S rRNA gene was performed by using 'universal' primers 515F (5′-GTGYCAGCMGCCGCGTAA-3′) and 806R (5′-GGACTACNVGGGTWTCTAA-3′) flanked with adapter and barcode sequences (Kuczynski et al., 2011). The PCR was carried out under the following conditions: 95 °C

**Table 1  Ingredient composition of the basal diet being fed for the broiler chickens used in this study.**

| Item | Amount (g/kg) |
|---|---|
| Ingredients | |
| Corn meal | 581.5 |
| Soybean meal | 335 |
| Soybean oil | 32.1 |
| Limestone | 13 |
| Dicalcium phosphate | 20.5 |
| L-lysine | 3.4 |
| DL-Methionine | 1.5 |
| Sodium chloride | 3 |
| Premix | 10 |
| Calculated nutrient levels | |
| Metabolizable energy (MJ/kg DM) | 12.08 |
| Crude protein (g/kg DM) | 19.25 |
| Calcium (g/kg DM) | 1.07 |
| Available phosphorus (g/kg DM) | 4.6 |
| Lysine (g/kg DM) | 12.6 |
| Methionine (g/kg DM) | 4.27 |
| Methionine + cysteine (g/kg DM) | 8.35 |

**Notes.**

DM, dry matter.

Premix provided per kilogram of diet: vitamin A (all-trans-retinyl acetate), 10,000 IU; vitamin D3 (cholecalciferol), 3,000 IU; vitamin E (all-rac-α-tocopherol), 30 IU; menadione, 1.3 mg; thiamin, 2.2 mg; riboflavin, 8 mg; nicotinamide, 40 mg; choline chloride, 400 mg; calcium pantothenate, 10 mg; pyridoxine HCl, 4 mg; biotin, 0.04 mg; folic acid, 1 mg; vitamin B12 (cobalamin), 0.013 mg; Fe (from ferrous sulphate), 80 mg; Cu (from copper sulphate), 8.0 mg; Mn (from manganese sulphate), 110 mg; Zn (from zinc oxide), 60 mg; I (from calcium iodate), 1.1 mg; Se (from sodium selenite), 0.3 mg.

for 5 min; 25 cycles of: 95 °C for 30 s, 56 °C for 45 s, 72 °C for 30 s; a final extension of 72 °C for 10 min, and then hold at 4 °C. The amplicons were cleaned by using AMPure XP beads (Beckman Coulter, Brea, CA, USA), and then normalized, pooled with the adapters and the dual indices using the Nextera XT Index Kit (Cat No.: FC-131-2001, Illumina, San Diego, CA, USA). A second PCR amplification with 5 cycles were executed with Nextera XT Index primers in following conditions: 95 °C for 4 min; 5 cycles of: 95 °C for 30 s, 55 °C for 40 s, 72 °C for 40 s; a final extension of 72 °C for 5 min, and then hold at 4 °C. The PCR products were cleaned up again with AMPure XP beads, and thus the sequencing libraries were established. The libraries were validated to the expected size of about 440 bp on a Bioanalyzer trace for the final library. The libraries were quantified using a Qubit 4.0 Fluorometer (Thermo Fisher Scientific, Waltham, MA, USA) according to the fluorometric quantification method using dsDNA binding dyes. The concentration of each DNA library was determined by an Agilent Technologies 2100 Bioanalyzer. For sequencing, the individual library was diluted for 4 nM, and aliquoted with 5 μl of diluted DNA, then pooled and sequenced on the Illumina Hiseq 4000 platform in paired-end (PE) technology at 2 × 250 nt using Illumina v2 kit (Illumina, San Diego, CA, USA) in Guhe Information Co., Ltd in Hanzhou, China.

## Metagenomic analysis

The raw reads from 16S rRNA sequencing were automatically input for quality control, trimming, demultiplexing of samples and then generated fastq output files. Afterwards, the reads were subjected to further proceeding by pipeline QIIME 2 (http://qiime.org/). Operational taxonomic units (OTUs), included de-replication, cluster, detection of chimera, were picked using Vsearch v1.11.1 based on a 97% 16S rRNA gene sequence identity level (*Rognes et al., 2016*). Taxonomic assignment of individual datasets was determined at several taxonomic levels: kingdom, phylum, class, order, family, genus, and species by using SILVA 128 (*Quast et al., 2013*). OTUs classified as chloroplasts or mitochondria were subsequently removed. The obtained sequences classified as bacteria and archaea were examined with BLAST (Basic Local Alignment Search Tool) (*Mount, 2007*).

Alpha diversity was calculated with QIIME, including indexes of observed species, chao1, shannon, simpson, and PD_whole_tree. Beta diversity was determined by using QIIME with the matrix of weighted and unweighted Unifrac distance. LEfSe analysis was performed by using linear discriminant analysis (LDA) to estimate the different size of the effect of abundance of each component (species), and to identify communities or species that had significant differences in the classification of the samples (*Segata et al., 2011*).

## Statistical analysis

The differences of data between EM-treated group and non-EM-treated group were analyzed by student $t$ test in SPSS 26.0 (IBM, USA). The replicate was defined as the experimental unit. Comparisons of parameters of growth performance across the groups were carried out by one-way analysis of variance (ANOVA) and significant differences among group means were determined using the least significant difference (LSD) test. The beta diversity indices were calculated based on the principal co-ordinates analysis (PCoA) method (*Quinn & K, 2002*). Kruskal-Walls test was used to identify the difference of alpha diversity indices and bacterial species which showed significant differences between different groups by R stats package. A $p$-value of <0.05 was set as the statistically significant level.

# RESULTS

## Effects of EM on growth performance of partridge shank broiler chicks

In this study, the diet with the addition of EM significantly increased the BW, ADG and decreased the FCR at the 10th ($P < 0.001$) and 20th ($P < 0.05$) day of the experiment when compared with the controls (Table 2). The BW gain for all the evaluated periods (day 1 to 10, day 1 to 20) was improved for chicks supplemented with EM. Similarly, ADG from day 1 to 10, from day 11 to 20 and overall period (day 1 to 20) were increased in EM-treated chicks. Moreover, FCR was decreased during day 1 to 10, and day 11 to 20, while no significance difference in all evaluated period. The EM addition had no significant impact on ADFI.

**Table 2** Growth performance of Partridge Shank broiler chickens fed diets supplemented with or without EM.

| Item | EM-treated boilers | Non-EM treated boilers | $p$–value (ANOVA) |
|---|---|---|---|
| BW, g | | | |
|     1 day | 145.70 ± 6.68 | 146.10 ± 5.28 | 0.846 |
|     10th day | 274.30 ± 10.80 | 254.60 ± 12.52 | 0.001[**] |
|     20th day | 563.40 ± 32.22 | 533.10 ± 19.55 | 0.020[*] |
| ADFI, g/day | | | |
|     1–10 days | 36.49 ± 8.54 | 35.47 ± 6.51 | 0.766 |
|     11–20 days | 56.74 ± 7.58 | 53.39 ± 7.07 | 0.321 |
|     1–20 days | 46.62 ± 13.02 | 44.43 ± 11.32 | 0.574 |
| ADG, g/day | | | |
|     1–10 days | 12.86 ± 1.27 | 10.85 ± 1.07 | 0.001[**] |
|     11–20 days | 28.91 ± 3.03 | 27.85 ± 1.85 | 0.048[*] |
|     1–20 days | 20.89 ± 3.38 | 19.33 ± 2.04 | 0.025[*] |
| FCR | | | |
|     1–10 days | 2.84 | 3.27 | 0.021[*] |
|     11–20 days | 1.96 | 1.92 | 0.049[*] |
|     1–20 days | 2.40 | 2.59 | 0.053 |

**Notes.**

BW, body weight; AVG, average; SD, Standard deviation; ADFI, average daily feed intake; ADG, average daily gain; FCR, feed conversion ratio.

[*]indicates $0.01 < p$ value $< 0.05$.

[**]indicates $0.01 < p$ value $< 0.001$.

## Effects of EM on the histomorphology of intestines of partridge shank broiler chicks

Diet with the supplementation of EM significantly increased the jejunal villus height ($P < 0.001$), but decreased the jejunal crypt depth ($P < 0.001$), and thus increased the ratio of jejunal villus height to crypt depth (VH/CD, $P < 0.001$). Furthermore, EM supplementation remarkably increased both ileal ($P < 0.001$) and cecal ($P < 0.001$) villus height and the ratio of VH/CD ($P < 0.001$), but decreased ileal ($P < 0.05$) and cecal ($P < 0.05$) crypt depth (Table 3).

## Microbial diversity of excreta and cecal microbiota of partridge shank broiler chicks

The samples ($n = 40$) were sequenced on an Illumina HiSeq 4000 platform and a total of 3,505,030 raw sequence reads were generated. After quality control, 3,234,992 (92.30%) clean reads were obtained, with an average of 80,874 clean sequences per sample (Table S2). Shannon, Simpson and Chao1 indices were employed to evaluate the alpha diversity within the sequence datasets based on the observed OTUs. Of the alpha diversity indices, no significant variation was observed between the comparable groups B0/EM0, B10/EM10, B20/EM20, and B20C/EM20C, indicating EM had limited influence on the alpha diversity indices (Table 4). The beta diversity among groups was presented on PCoA to distinguish the microbial communities (Fig. 1). The results revealed that the microbial communities in cecal contents showed a striking distinctness with that in excreta. Clusters of excreta

**Table 3  The villus height and crypt depth of intestine in chickens between EM-treated and non-EM-treated negative control groups.**

| Item | Intestinal section | Average length ± standard deviation, μm | | *p*-value (ANOVA) |
|------|--------------------|-------------------|----------------|-------------------|
| | | **EM-treated** | **Control** | |
| Villus height (VH) | Jejunum | 575.35 ± 59.28 | 427.28 ± 52.80 | 0.000*** |
| | Ileum | 520.13 ± 42.93 | 342.79 ± 22.47 | 0.000*** |
| | Cecum | 82.83 ± 17.32 | 50.44 ± 9.27 | 0.000*** |
| Crypt depth (CD) | Jejunum | 39.55 ± 10.46 | 52.42 ± 11.88 | 0.001** |
| | Ileum | 35.06 ± 10.14 | 42.03 ± 9.88 | 0.034* |
| | Cecum | 19.01 ± 2.91 | 22.42 ± 4.86 | 0.011* |
| VH/CD value | Jejunum | 15.45 ± 4.12 | 8.51 ± 2.06 | 0.000*** |
| | Ileum | 16.16 ± 5.69 | 8.64 ± 2.30 | 0.000*** |
| | Cecum | 4.46 ± 1.14 | 2.39 ± 0.79 | 0.000*** |

Notes.
*indicates $0.01 < p$ value $< 0.05$.
**indicates $0.001 < p$ value $< 0.01$.
***indicates $p$ value $< 0.001$.

**Table 4  Microbiota alpha diversity among groups of chickens by Kruskal–Walls test.**

| Group | Shannon | Simpson | Chao1 | Ace | Goods_coverage |
|-------|---------|---------|-------|-----|----------------|
| B0 | 3.41 ± 1.33 | 0.696 ± 0.26 | 392.94 ± 110.63 | 383.08 ± 110.63 | 1.00 ± 0.00 |
| EM0 | 3.57 ± 0.96 | 0.76 ± 0.12 | 433.23 ± 188.59 | 433.07 ± 186.16 | 1.00 ± 0.00 |
| *p*-value | 0.83 | 1.00 | 0.69 | 0.62 | 1.00 |
| B10 | 3.51 ± 1.09 | 0.74 ± 0.10 | 538.89 ± 209.50 | 536.24 ± 203.14 | 1.00 ± 0.00 |
| EM10 | 3.17 ± 1.44 | 0.69 ± 0.25 | 584.54 ± 201.78 | 563.80 ± 198.39 | 1.00 ± 0.00 |
| *p*-value | 0.69 | 1.00 | 0.69 | 0.83 | 1.00 |
| B20 | 4.86 ± 1.89 | 0.84 ± 0.17 | 899.23 ± 201.02 | 897.80 ± 190.29 | 1.00 ± 0.00 |
| EM20 | 4.36 ± 1.51 | 0.86 ± 0.08 | 857.33 ± 247.88 | 853.84 ± 250.80 | 1.00 ± 0.00 |
| *p*-value | 0.55 | 1.00 | 0.84 | 0.76 | 1.00 |
| B20C | 6.56 ± 0.42 | 0.96 ± 0.02 | 697.89 ± 398.22 | 689.15 ± 388.87 | 1.00 ± 0.00 |
| EM20C | 6.41 ± 0.63 | 0.96 ± 0.03 | 1061.42 ± 163.78 | 1054.42 ± 159.89 | 1.00 ± 0.00 |
| *p*-value | 1.00 | 0.84 | 0.15 | 0.09 | 1.00 |

microbiota were superimposed over the PCoA analysis and represented the differences among the groups.

## Comparison of microbial communities of excreta microbiota between EM-treated and non-EM-treated chicks

In the composition analysis at the phylum level, *Firmicutes* accounted for the most of the relative abundance (63.24%~92.63%), followed by *Proteobacteria* (0.62%~23.94%), *Bacteroidetes* (0.80%~7.85%), *Actinobacteria* (0.06%~13.69%) and others. With increasing age, the abundance of *Firmicutes* tended to decrease and the abundance of *Proteobacteria*, *Bacteroidetes* and *Actinomycete* tended to increase (Fig. 2 and Table S3). At the genus level, *Lactobacillus* had the highest relative abundance in the excreta samples (33.26%~78.03%),

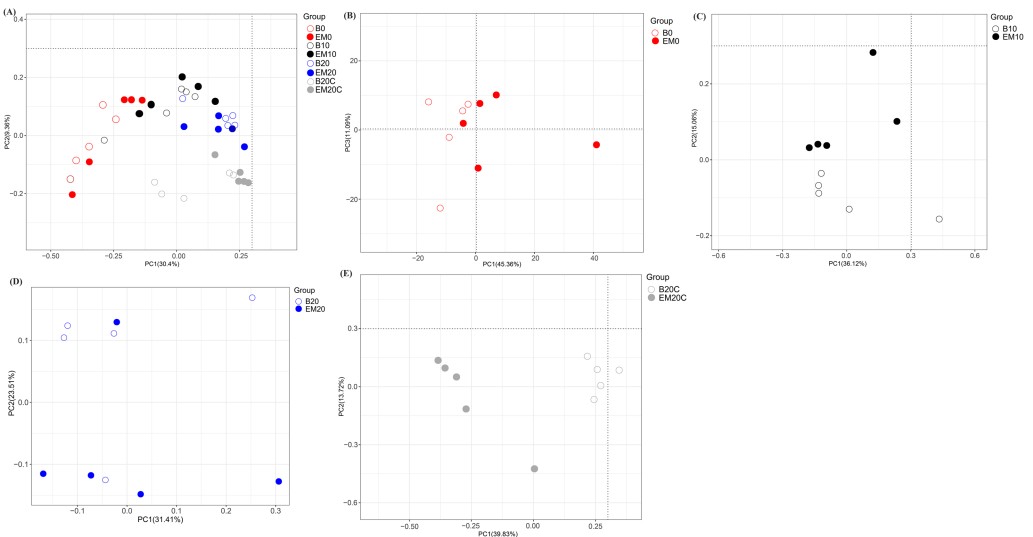

**Figure 1** Principal component analysis (PCoA) based on the sequences from all samples tested (A), excreta samples from the 0 (B), 10th (C), and 20th (D) experimental day, and cecal content samples from the 20th experimental day (E).

followed by *Streptococcus* (0.01%∼19.07%), *Enterococcus* (0.16%∼20.94%), and *Bacteroides* (0.46%∼5.27%). Similarly, the abundances of *Lactobacillus* in feces in both EM-treated and non-EM-treated broiler chicks were reduced, while the abundances in EM-treated chicks were higher than that in non-EM-treatedled chicks. However, in cecum samples, an unclassified genus from the family *Lachnospiraceae* accounted for the most of the relative abundance (24.02%∼36.41%), followed by unclassified genus from the order *Clostridiales* (20.32%∼23.40%), unclassified genus in the family *Lachnospiraceae* (6.50%∼7.30%), and *Ruminococcus* (2.82%∼4.63%) (Fig. 3 and Table S4).

## Gut microbiota landscape of non-EM-treated chicks

To explore the gut microbiota landscape of the boiler chicks in non-EM-treated group, ANOVA test was performed. At the phylum level, the abundances of *Firmicutes* were significantly decreased (90.53% to 63.24%) from B0 (10d age) to B20 (30d age). While the abundances of *Euryarchaeota*, *Synergistetes*, *Verrucomicrobia*, and *Actinobacteria* were significantly increased as the chicks grew up (Table 5 and Fig. S1A). At the genus level, abundances of *Prevotella*, *Coprococcus*, *Desulfovibrio*, *Gallibacterium*, and *Acinetobacter* tended to be increased from age 10d to 30d (Fig. S1B).

## Gut microbiota landscape of EM-treated chicks

Among the EM-treated partridge shank broiler chicks, four kinds of gut bacteria at the phylum level were significantly different among growth stages of EM0, EM10 and EM20. Bacteria in *Proteobacteria*, *Synergistetes, and WPS-2* were significantly increased from EM0 to EM20 (Table 6 and Fig. S2A). At the genus level, the abundances of *Methanobrevibacter*, *Enterococcus*, *Streptococcus,* and *Gallibacterium* were significantly augmented in EM10,

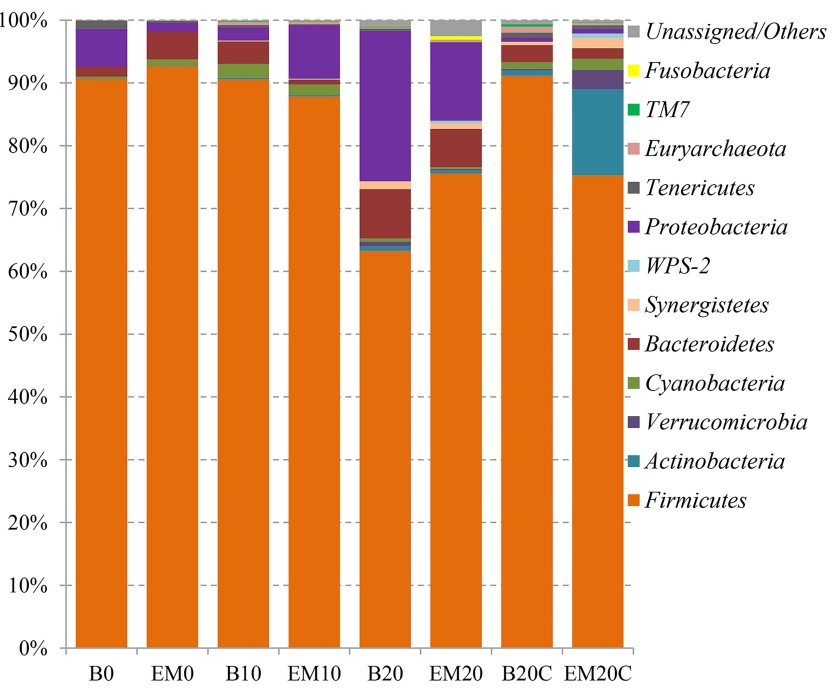

**Figure 2** **Gut microbial composition at phylum-level.**

while decreased in EM20. The abundances of *Faecalibacterium* and *Megamonas* were reduced during the time of EM supplementation (Fig. S2B).

## Comparison of excretal and cecal microbiota between EM-treated and non-EM-treated chicks

To address the impacts of EM on the structure and abundance of microbiota in feces and cecal contents, the abundances of fecal and cecal bacteria in chicks at the end of the experiment were analyzed. For the excretal microbiota, the abundances of two bacteria *TM7* ($P < 0.01$) and *Tenericutes* ($P < 0.01$) at the phylum level and one at the genus level *Acinetobacte* ($P < 0.05$) were reduced in group EM20 when compared with that in group B20 (Table 7 and Fig. S3). As the normal structure of bacterial communities in ceca was very different from that in excreta, the changes of cecal microbiota in EM-treated broiler chicks were different. When compared with the control group, the abundances of *Firmicutes* ($P < 0.001$), *Euryarchaeota* ($P < 0.05$), and *Ruminococcus* ($P < 0.05$) were significantly reduced, while the abundances of *Actinobacteria* ($P < 0.001$) and *WPS-2* ($P < 0.001$) were significantly increased (Table 8 and Fig. S4).

## DISCUSSION

As reported from previous studies, supplementation of probiotics in feed could improve the feed intake, weight gain and feed efficiency in broilers (*Waititu et al., 2014*; *Qorbanpour et al., 2018*; *Jha et al., 2020*). Therefore, in order to enhance the growth rate, maintain intestinal integrity, and improve the overall health status of birds in intensive production

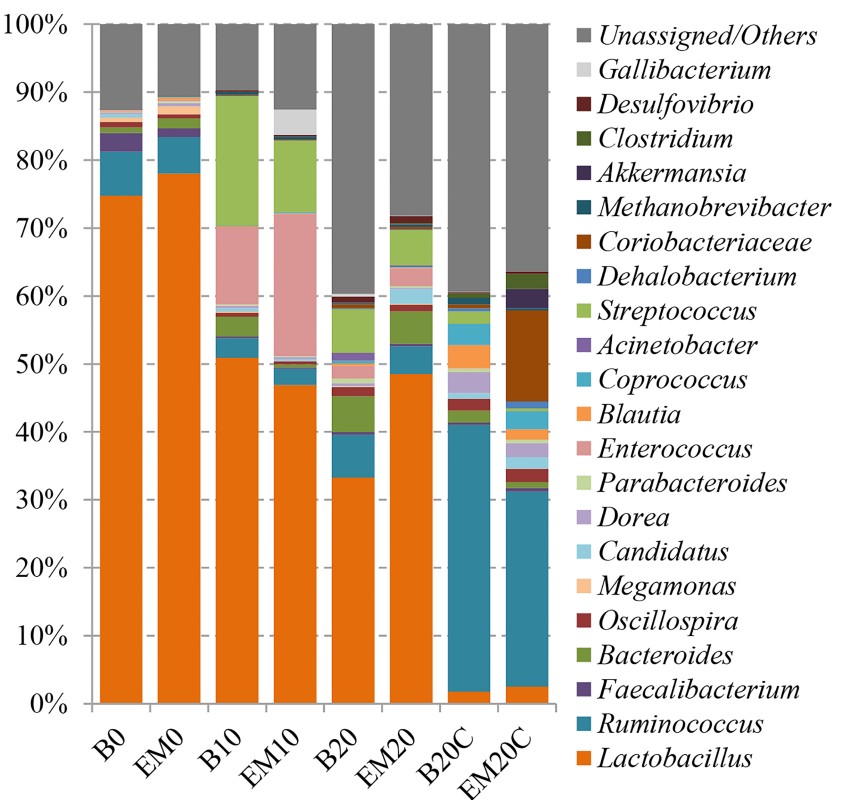

**Figure 3** Gut microbial composition at genus-level.

**Table 5** Abundance differences of bacteria among the bird gut in non-EM-treated negative control group.

| Bacterial name | Average abundance, % | | | ANOVA test $p$ value | Significance |
|---|---|---|---|---|---|
| | B0 | B10 | B20 | | |
| Firmicutes | 90.5325 | 90.5611 | 63.2401 | 0.032 | *** |
| Euryarchaeota | 0.0003 | 0.5090 | 0.1959 | 0.000 | *** |
| Synergistetes | 0.0000 | 0.1370 | 1.2271 | 0.000 | *** |
| Verrucomicrobia | 0.0001 | 0.0222 | 0.7400 | 0.000 | *** |
| Actinobacteria | 0.0000 | 0.0003 | 0.0008 | 0.005 | *** |
| Methanobrevibacter | 0.0003 | 0.4394 | 0.1830 | 0.000 | *** |
| Prevotella | 0.0045 | 0.0701 | 0.2755 | 0.000 | *** |
| Streptococcus | 0.0116 | 19.0748 | 6.4261 | 0.000 | *** |
| Coprococcus | 0.0383 | 0.0906 | 0.4035 | 0.000 | *** |
| Desulfovibrio | 0.0056 | 0.0047 | 0.0640 | 0.000 | *** |
| Gallibacterium | 0.0000 | 0.0128 | 0.3822 | 0.000 | *** |
| Acinetobacter | 0.0252 | 0.0022 | 1.1387 | 0.000 | *** |

**Table 6  Abundance differences of bacteria among the gut of EM-treated broiler chickens.**

| Bacterial name | Average abundance, % | | | Variation test p value | Significance |
|---|---|---|---|---|---|
| | EM0 | EM10 | EM20 | | |
| *Euryarchaeota* | 0.0001 | 0.4001 | 0.3005 | 0.0000 | *** |
| *Proteobacteria* | 1.4001 | 8.5000 | 12.4002 | 0.0040 | *** |
| *Synergistetes* | 0.0000 | 0.0002 | 0.9000 | 0.0000 | *** |
| *WPS-2* | 0.0000 | 0.1000 | 0.4002 | 0.0000 | *** |
| *Methanobrevibacter* | 0.0000 | 0.4000 | 0.3000 | 0.0000 | *** |
| *Enterococcus* | 0.3001 | 20.9002 | 2.6001 | 0.0000 | *** |
| *Streptococcus* | 0.0001 | 10.5001 | 5.2003 | 0.0000 | *** |
| *Faecalibacterium* | 1.3000 | 0.2000 | 0.3000 | 0.6070 | NA |
| *Megamonas* | 1.2001 | 0.0001 | 0.1001 | 0.0000 | *** |
| *Desulfovibrio* | 0.0001 | 0.3000 | 1.0000 | 0.0000 | *** |
| *Gallibacterium* | 0.0000 | 3.7000 | 0.1000 | 0.0000 | *** |

**Notes.**
*** indicates $p$ value < 0.001.

**Table 7  Abundance differences of bacteria in feces between EM-treated and non-EM-treated negative control Partridge Shank broiler chickens.**

| Bacterial name | Average abundance, % | | Variation test p value | Significance |
|---|---|---|---|---|
| | B20 | EM20 | | |
| *TM7* | 0.2001 | 0.0000 | 0.0021 | ** |
| *Tenericutes* | 0.4001 | 0.1002 | 0.0021 | ** |
| *Acinetobacter* | 1.1000 | 0.2000 | 0.0350 | * |

**Notes.**
* indicates $0.01 < p$ value < 0.05.
** indicates $0.001 < p$ value < 0.01.

**Table 8  Abundance differences of bacteria in cecal contents between EM-treated and non-EM-treated negative control Partridge Shank broiler chickens.**

| Bacterial name | Average abundance, % | | Variation test p value | Significance |
|---|---|---|---|---|
| | B20C | EM20C | | |
| *Euryarchaeota* | 1.0003 | 0.3001 | 0.0431 | * |
| *Actinobacteria* | 0.8001 | 13.7001 | 0.0000 | *** |
| *Firmicutes* | 91.1000 | 75.3000 | 0.0051 | ** |
| *WPS-2* | 0.0002 | 0.7001 | 0.0000 | *** |
| *Ruminococcus* | 15.8001 | 4.1001 | 0.0350 | * |

**Notes.**
* indicates $0.01 < p$ value < 0.05.
** indicates $0.001 < p$ value < 0.01.
*** indicates $p$ value < 0.001.

conditions, the use of probiotic preparations as a supplement is a common practice in poultry production (*Wondmeneh, Getachew & Dessie, 2011*). In this study, an EM mixture containing multiple species of bacteria, of which most are naturally existing beneficial microorganisms, including both oxybiotic and anaerobic microbes, was applied to evaluate the effects on the growth performance, intestinal histomorphology, and gut microbiota of partridge shank broiler chicks. Researchers have reported that probiotics had positive effects on BW and ADG of animals (*Huang et al., 2019*; *Tao et al., 2021*). The functional inconsistency of probiotics among these studies, including the present study, might attribute to the type and dosage of probiotics being used, and the breeds of the broilers as well.

In this study, positive effects of the EM supplementation on BW, ADG and FCR were found in partridge shank broiler chicks. The BW gain and ADG were significantly higher in EM-fed chicks than that in control chicks both at the first phase (0–10th day) and the second phase (11–20th day) of the experiment. However, the ADFI showed no difference between the EM-fed and non-EM-treated control chicks in both phases, which indicated that the EM supplementation could improve the feed conversion efficiency and led to the decrease of FCR. These findings agree with previous studies regarding the beneficial effects of EM and probiotics on the growth performance and gut health of partridge shank broiler chicks (*Chantsawang & Watcharangkul, 1999*; *Safalaoh, 2006*; *Alkhalf, Alhaj & Al-Homidan, 2010*; *Xu et al., 2014*; *Fazelnia et al., 2021*). *Chantsawang & Watcharangkul (1999)* evaluated the effects of EM supplementation on 4 different types of poultry, and it was found that EM additive could significantly increase breast percentage in Muscovy duck, and decrease ash content of breast meat in Arbor Acers broiler chickens. *Safalaoh (2006)* demonstrated that the addition of EM in diet significantly increased BW gains (2094 ± 11 g) and ADG compared to broilers on the control diet (2057 ± 15 g) during 1–42 days of age. However, on the other hand, there are some reports which state that probiotics or EM had no role on the growth performance and mortality in broilers. It has been found that the addition of probiotics or prebiotics in broiler diet reduced feed intake (*Mokhtari et al., 2010*; *Chen et al., 2015*). Others have demonstrated that the use of probiotics in broiler diet did not affect FCR (*Sarangi et al., 2016*). In addition, it was found that the weight gain was not affected by supplementation of probiotics (*Yousefi & Karkoodi, 2007*). The inconsistent results of aforementioned studies might attribute to multiple factors, including probiotic types and dose being used, and other potential element(s) (*Kabir, 2009*; *Sohail et al., 2012*).

In this study, the EM addition positively influenced the histomorphological characteristics of the broilers' intestine. Histological observation indicated the supplementation of EM increased the height of intestinal villi in jejunum, ileum and cecum. The structure of intestinal villi were covered with the intestinal epithelium, under which there was a continuous cell layer of myofibroblasts that could regulate the epithelial renewal and defense processes (*Ackermann, Nowicki & Sarnecka-Keller, 1974*). Furthermore, EM also decreased the intestinal crypt depth and increased VH/CD ratio of broilers. Crypts are associated with the proliferation of epithelial cells by producing defensins and dendocrine substances (*Manning & Gibson, 2004*). It has been demonstrated that probiotics *Saccharomyces boulardii* and *Bacillus cereus* had beneficial effects on

the epithelial structure and cryptic morphology (*Baum et al., 2002*). *Award, Ghareeb & Bohm (2009)* found that addition of probiotics composed of *Lactobacillus salivarius* and *Lactobacillus reuteri* in feed and found that probiotics significantly increased the BW, average daily weight gain, and improved the villus integrity in small intestines, increased the VH/CD ratio in duodenum in broilers. The EM used in this study contained multiple probiotic bacteria, such as *Lactobacillus* (abundance of 84% ± 12%, Table S1) and *Bacillus* (0.09% ± 0.11%), which might benefit the villus and cryptic morphology and then promote the intestinal health.

Probiotics were suitable for domestic animals, because they can inhibit the growth of pathogenic bacteria and promote the growth of beneficial bacteria by producing different metabolites and thus improve the gut microecological environment (*Cummings & Kong, 2004*; *Attia et al., 2013*; *Sun et al., 2019*; *Robinson et al., 2020*). Similar results were observed in the present study. Although the abundances of *Lactobacillus* were reduced in the chicks as they got older, the abundance of *Lactobacillus* in EM-treated partridge shank broiler chicks were elevated when compared with that in non-EM-treated broiler chicks. The abundance of *Acinetobacter* was significantly lower in EM20 compared to that of B20. It is well known that most of the members of *Acinetobacter* enteropathogenic and can cause infections (*Michalopoulos & Falagas, 2010*). The commonly encountered pathogenic or zoonotic bacteria affecting birds, including *E. coli*, *Streptococcus*, and *Clostridium* were slightly reduced in the gut of EM-treated partridge shank broiler chicks. Together, the supplementation of EM in feed could ameliorate the community and structure of the intestinal microbiota of partridge shank broiler chicks.

## CONCLUSIONS

In this study, we observed that the feed supplemented with EM could increase the body weight and average daily gain, and reduced feed conversion ratio, enhance intestinal integrity, and balance the gut microflora of partridge shank broiler chicks. The findings might provide an alternative to improve the growth performance and the gut health of partridge shank broiler chicks.

## ACKNOWLEDGEMENTS

We thank Prof. Nanhui Chen for providing the effective microorganism suspensions.

### Funding

This work was funded by the National Natural Science Foundation of China (31960711) and the Natural Science Foundation of Jiangxi Province (20202BABL215024). The funders had no role in study design, data collection and analysis, decision to publish, or preparation of the manuscript.

## Grant Disclosures

The following grant information was disclosed by the authors:
The National Natural Science Foundation of China: 31960711.
The Natural Science Foundation of Jiangxi Province: 20202BABL215024.

## Competing Interests

Bin Zhu is employed by Jiangxi Red animal health products Co., LTD; and Longrui Li is employed by Jiangxi Newtoldhow animal pharmaceutical Co., LTD.

## Author Contributions

- Yizhe Ye and Zhiquan Li performed the experiments, prepared figures and/or tables, authored or reviewed drafts of the paper, and approved the final draft.
- Ping Wang, Bin Zhu, Min Zhao and Dongyan Huang performed the experiments, prepared figures and/or tables, and approved the final draft.
- Yu Ye, Longrui Li and Gen Wan analyzed the data, prepared figures and/or tables, and approved the final draft.
- Zhen Ding analyzed the data, authored or reviewed drafts of the paper, and approved the final draft.
- Qiong Wu analyzed the data, authored or reviewed drafts of the paper, project administration, and approved the final draft.
- Deping Song conceived and designed the experiments, authored or reviewed drafts of the paper, project administration, and approved the final draft.
- Yuxin Tang conceived and designed the experiments, authored or reviewed drafts of the paper, and approved the final draft.

## Animal Ethics

The following information was supplied relating to ethical approvals (i.e., approving body and any reference numbers):

The Office of Animal Care and Use of Jiangxi Agricultural University provided full approval for this research (JXAU-LL-20190022).

## Data Availability

The sequence data is available at the Short Reads Archive (SRA) database at NCBI: PRJNA629019.

Raw data are available in the Supplemental File.

## Supplemental Information

Supplemental information for this article can be found online at http://dx.doi.org/10.7717/peerj.12538#supplemental-information.

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
