# Peer review of "Effects of probiotic supplements on growth performance and intestinal microbiota of partridge shank broiler chicks"

_PeerJ, doi:10.7717/peerj.12538_

## Round 0.1 · original submission · Major Revisions

Three reviewers have now provided their opinions and comments on the manuscript. Please provide responses and explanations to the reviewers and make the changes in the manuscript where appropriate. If you disagree with any of the reviewers’ requests, please provide your reasoning in the rebuttal letter. It is my opinion that no additional in vivo experiments are needed for considering the manuscript for publication.

Reviewer 1 ·

Basic reporting

The article must be written in English and must use clear, unambiguous, technically correct text. The article must conform to professional standards of courtesy and expression.

Experimental design

The design was poor. All the study were just 2 groups, without positive control group. How to compare the advantage of your EM without the ruler. Hence, the data was unconfident.

Validity of the findings

In section of Background: “to our knowledge, there is no report of EM on the growth performance and the modulation in GM of Partridge Shank broiler chickens: were wrong. EM have been used in poultry chicks, also in Partridge Shank broiler chickens. The authors must read more references on direct supplementary probiotic feed.

Additional comments

Although this work was carried out with efforts. The related research was very heavy. And the innovation of this study was few. There was no corrected analysis on the data of growth performance and intestinal micro structure. The most issue was the uncorrected group design cause the inappropriate data.

Reviewer 2 ·

Basic reporting

The manuscript describes a study to investigate the efficacy of probiotic supplementation on growth performance and intestinal microbiota of partridge shank broiler chickens between 10-30 days of age. The novelty of the manuscript is fair, but there are several problems. First, the author's description of significance is not particularly clear, including the writing and description of the P value. When the comparison between the test group and the control group is significant in a certain parameter, the P value needs to be added to the end of the sentence. For example, line 258. On the contrary, there is no need to add P value. And in general, there is no need to report or discuss non-significant results because they do not contribute to the research results.

Experimental design

.

Validity of the findings

.

Additional comments

Line 31-33: There is only one in the experimental group. If you can do two more experiments with different EM additions, the discussion of the results may be richer and have practical guiding significance.
Line 78-79: The connection between these two paragraphs is a bit awkward, you may need to add one sentence.
Line 202-203: Do not use the first paragraph for each of the results subsections to indicate what was done (as this is already included in materials and methods section) and a caption of the representative table. Instead, directly and concisely report the results and refer to the table as appropriate in the text.
Line 205: The P value should be presented P < 0.05, P < 0.01 or P < 0.001. Please change this consistently within the manuscript. Moreover, non-significant results do not need to be reported.
Line 216-220: Same as the problem presented in line 202-203.
Line 222-223: Delete the sentence “However, the indices of Shannon, Simpson, Chao1 and Ace tended to be higher as the broilers grew up”
Line 264-275: No P value was presented in the paragraph.
Line 285: Whether the word “Partridge Shank” need to be italicized, please unify it.
Line 290: “(Sohail et al., 2012)” should be presented as “(2012)”.
Line 290-291: Inconsistent statements and references.
The latest references should be added in your manuscript. You can check the work in
1. Production performance, egg quality, plasma biochemical constituents and lipid metabolites of aged laying hens supplemented with incomplete degradation products of galactomannan
2. A sustainable process for procuring biologically active fractions of high-purity xylooligosaccharides and water-soluble lignin from Moso bamboo prehydrolyzate

·

Basic reporting

The research background is introduced in detail in the preface, being there are few studies about the impact of EM on local chicken breeds, so this article was carried out to evaluate the effects of EM on the growth performance, gut health and microbiota of Partridge Shank broiler chickens.The reason and basis are sufficient, and the previous research literature references are also included. The structure, figures and tables are put in correctly. All appropriate raw data have been made available. The submission include the results relevant to the hypothesis that the EM would improve the structure and composition of gut microbiota of EM-treated broiler chickens, perhaps via a mechanism of inhibiting the colonization of bad bacteria. The language is clear and intelligible, but not professional. Some examples where the language could be improved include lines 71,72,79,80, 81, the current phrasing makes comprehension difficult . I suggest you have a colleague who is proficient in English and familiar with the subject matter review your manuscript.

Experimental design

The article aimed to evaluate the effects of a basal diet with the supplementation of an EM mixture on the growth performance and gut health in Partridge Shank broiler chickens. The experiment was performed to a high technical and ethical standard.
All procedures involving live animals were verified and approved by the Office of Animal Care and Use of Jiangxi Agricultural University (protocol number JXAU-LL-2020-0018). Methods were described with sufficient detail and information to be reproducible by another investigator. But the number of replication and the replication per are too few. According the experimental design, 6 replications are necessary, and 30-50 chickens per replication are more suitable.

Validity of the findings

All underling data have been provided, and they are robust, statistically sound and controlled. Some meaningful research results are obtained, i.e when compared with the control group, in cecal the abundances of Firmicutes, Euryarchaeota, and Ruminococcus were significantly reduced, while the abundances of Actinobacteria and WPS-2 were significantly increased. But in the discussion, there is no explanation for the significance of this change in intestinal microbiota. Please add it. Moreover, the microbiome in feces is very different from the intestinal microbiome, the study of intestinal microbiome is more meaningful than fecal microbiome. Why does this study focus on fecal microbiome? The conclusions are not well stated, it is too long, should be accurately summarized.

Additional comments

1. Line 28 Methods: It is too simply, we don't know which indicators have been sampled and tested.
2. Line 37 results: the results of gut microbiota,come form feces or cecal? We donot know.
3. Line 81: According to reports, local breeds of broiler chicken account for 46.52% of broiler slaughter in China, 2017, and this proportion will continue to increase in 2018 (Zhao et al., 2019), are there any up-to-date references? It's too old.
4. Line 108: vaccines against Marek’s disease, Newcastle disease, and Infectious bursal disease,the specific immunization date should be given.
5. Partridge shank chicken, a local broiler breed in China, using NRC (1994) standard is not suitable, do you think?
6. line 121: basal diet supplemented with 2.5 x 109 colony-forming unit (cfu) of EM per bird/day for 20 days, how to do it, add it to the feed?
7. Line 127: Then the feed intake was calculated from the differences between the offered and residual feed and was used to calculate average daily intake (ADFI), average daily gain(ADG), and feed/gain ratio (F/G). How to calculate the ADG through the feed intake?
8. Line 132: excreta at the 1st, designated as EM0, why not designated as EM1? Same as B0.
9. Line 135: Fecal samples and cecal contents from 3 chickens in each coop were pooled as one test specimen. Since five chickens in each group were sampled, how to deal with remaining two?
10. Line 153: What is the composition of the PCR reaction system?Please in detail.
11. Line 158: with Nextera XT Index primers, what is primers sequence?
12. Line 220:“No significant variation was observed between”, “was “ should be “were”.
13. The result is sometimes in the past tense and sometimes in the present tense. Such as Line224 and line 231.
14. Line 290: Controversially, Falaki et al observed that the feed intake was decreased by the addition of prebiotics (Sarangi et al., 2016). Prebiotics are not probiotics, it is inappropriate to compare the two together.
15. “ Chantsawang, 1999; Wondmeneh et al., 2015”, these two references were not found in the reference.

---

## Round 0.2 · Minor Revisions

Thank you for revising your manuscript and addressing the Reviewers' comments. Two Reviewers have reviewed and approved the revised manuscript. However, there are a few aspects that were pointed out by the Section Editor and which need to be addressed before the manuscript can be accepted for publication:

1) The paper states: "Fecal samples and cecal contents from 3 chicks in each coop were pooled as one test specimen." Pooling samples for microbiota analysis is not a widely accepted practice and it takes away the individual variation effect in the study.

2) Reviewer 1 stated that "EM have been used in poultry chicks, also in Partridge Shank broiler chickens. The authors should add more references on direct supplementary probiotic feed". Please add relevant background about the state-of-the-art of this field and the specific topic of this manuscript.

3) Please address the concern of Reviewer 3 about the discussion. The results of the study should be discussed in the context of the published literature and current knowledge about the EM effects on modulation of GM.

In addition, please correct the writing of "partridge shank" and "chicks" which should not be in Italics as written in several places of the current manuscript.

Reviewer 2 ·

Basic reporting

It is ok now.

Experimental design

It is ok now.

Validity of the findings

It is ok now.

Additional comments

The authors have revised the manuscript according to the reviewers. It can be accepted now

·

Basic reporting

no comment

Experimental design

no comment

Validity of the findings

no comment

Additional comments

The discussion is not deep enough. However, all the amendments mentioned have been revised. It is ok.

---

## Round 0.3 · Minor Revisions

Thank you for including additional background and discussion in the manuscript and clarifying the experimental aspects. In the attached manuscript pdf I have provided some language edits, mainly in the newly added sections during the last revision. Please correct the language and also provide additional details about the source of the effective microorganisms (EM) used in the study. I believe after these corrections the manuscript will be suitable for publication.

---

## Round 0.4 · accepted · Accept

Thank you for addressing all the comments in your revised manuscript.